Predicting mortality in geriatric patients with peptic ulcer bleeding: a retrospective comparative study of four scoring systems

Aydin Omerul Faruk 1
Tatlıparmak Ali Cankut alicankut.tatliparmak@uskudar.edu.tr 2
1 Department of Emergency Medicine, İstanbul Yeni Yüzyıl University , İstanbul , Turkey
2 Department of Emergency Medicine, Üsküdar University , İstanbul , Turkey
Samuel Stephen
Electronic publication date: 2025 Mar 17
Publication date: 2025
Volume: 13
Electronic Location ID: e19090
Received 2024 Dec 10; Accepted 2025 Feb 11
Copyright: ©2025 Aydin and Tatlıparmak
Copyright year: 2025
Copyright holder: Aydin and Tatlıparmak
License: This is an open access article distributed under the terms of the Creative Commons Attribution License, which permits unrestricted use, distribution, reproduction and adaptation in any medium and for any purpose provided that it is properly attributed. For attribution, the original author(s), title, publication source (PeerJ) and either DOI or URL of the article must be cited.
License URL: https://creativecommons.org/licenses/by/4.0/

Keywords: Upper gastrointestinal bleeding, Geriatrics, Mortality

Funding: The authors received no funding for this work.

==============================
Background

Peptic ulcer bleeding (PUB) is a significant cause of morbidity and mortality, especially in geriatric patients. Risk stratification tools such as AIMS65, Glasgow Blatchford Score (GBS), T-score, and Age, Blood tests, and Comorbidities (ABC) score are frequently used to predict outcomes in PUB patients. This study aims to compare the predictive performance of these four scoring systems in geriatric patients with PUB.

Methods

This retrospective cohort study included patients aged 65 years and older who were diagnosed with PUB between January 1, 2019, and January 1, 2024, in a tertiary care hospital. Data collected included demographic information, clinical presentation, laboratory results, and comorbidities. AIMS65, GBS, T-Score, and ABC score were calculated for each patient. The primary outcome was in-hospital mortality.

Results

A total of 315 patients were included in the study, with an overall in-hospital mortality rate of 7.9%. AIMS65 had the highest area under the curve (area under the receiver operating characteristic curve (AUROC): 0.829), followed by the ABC score (AUROC: 0.775). The GBS (AUROC: 0.694) and T-score (AUROC: 0.526) demonstrated lower predictive performance. Pairwise comparisons showed a statistically significant difference between the AIMS65 and GBS (p = 0.0214). AIMS65 was the most accurate predictor of in-hospital mortality in geriatric PUB patients.

Conclusion

The AIMS65 and ABC scoring systems are more effective in predicting in-hospital mortality in geriatric patients with PUB compared to the GBS and T-Score. Implementing these tools in clinical practice could improve risk stratification and decision-making processes in managing high-risk elderly patients.

Introduction

Upper gastrointestinal bleeding (UGIB) is a common clinical condition in emergency departments (ED), carrying a significant risk of morbidity and mortality. Although it can affect all age groups, it tends to have more severe outcomes, particularly in the geriatric population (Stolow et al., 2021; Dogru et al., 2022; Di Gioia et al., 2024). In elderly individuals, delays in clinical recognition of bleeding, the presence of comorbidities, and age-related physiological changes contribute to the complexity of UGIB management. As a result of these challenges, hospitalization and mortality rates in geriatric patients are markedly higher compared to younger populations (El-Dallal et al., 2022; Li et al., 2023). Therefore, early risk assessment, clinical management, and prognosis are critically important for this high-risk patient group.

In recent years, the use of risk stratification systems for UGIB patients has been recommended by international guidelines (Laine et al., 2021; Gralnek et al., 2022). These systems contribute to the rapid and effective determination of clinical management plans for patients. Specifically, the scoring systems used in the management of patients presenting with UGIB play a critical role in predicting mortality, rebleeding, and the need for intervention (Kılıç et al., 2022).

Currently, scoring systems such as AIMS65, Glasgow Blatchford Score (GBS), T-score, and Age, Blood tests, and Comorbidities (ABC) score are widely used to estimate the risk of mortality associated with UGIB (Ebrahimi Bakhtavar et al., 2017; Li et al., 2022). While these scoring systems provide valuable contributions to clinical practice, conducting an analysis specific to the geriatric patient population would allow for a better understanding of their effectiveness across different age groups. Age-related physiological changes, particularly the deterioration of hemodynamic stability, slower tissue healing, and the increased burden of comorbidities, are thought to affect the accuracy of risk assessment in this population (Pilotto et al., 2011). Therefore, clearly demonstrating the performance of these systems in geriatric patients could contribute to more effective planning of risk management and clinical decision-making processes in the future.

In this study, we aimed to compare the effectiveness of the AIMS65, GBS, T-score, and ABC score risk assessment systems in predicting in-hospital mortality following peptic ulcer bleeding (PUB) in geriatric patients.

Materials & Methods

This study was conducted with the approval of the Yeni Yüzyıl University Clinical Research Ethics Committee (Date: 04.11.2024, Decision No: 2024/11-1358). The study was carried out in accordance with the ethical principles of the Declaration of Helsinki. Written informed consent is obtained from each participant or their next of kin. This retrospective study was conducted on patients aged 65 and over who were diagnosed with PUB in the emergency department of a tertiary healthcare institution between January 1, 2019, and January 1, 2024.

The inclusion criteria were being diagnosed with PUB, being 65 years or older, and having a complete dataset during hospitalization. Geriatric patients were defined as individuals aged 65 years and older, consistent with widely accepted clinical guidelines and research conventions. The age of 65 is recognized as the threshold for geriatric classification due to its historical use in retirement and healthcare policies, as well as its prevalence in medical research definitions (Sabharwal et al., 2015; United Nations, 2020). The exclusion criteria included having a diagnosis other than PUB, being under 65 years of age, and having incomplete data that would prevent the calculation of the scoring systems. UGIB was defined as hemorrhage originating from the mouth to the ligament of Treitz, encompassing bleeding from the esophagus, stomach, or duodenum (Kim et al., 2020). The diagnosis was established based on clinical manifestations such as hematemesis (vomiting of blood), melena (black, tarry stools), or hematochezia (passage of fresh blood per rectum) in cases of significant bleeding. Laboratory findings, including decreased hemoglobin and hematocrit levels, supported the clinical diagnosis. Definitive confirmation was obtained through esophagogastroduodenoscopy (EGD), which identified sources of bleeding such as peptic ulcers, erosions, or varices (Wilkins, Wheeler & Carpenter, 2020). PUB is defined as bleeding originating from peptic ulcers in the stomach or duodenum, confirmed by EGD. Clinical presentations and laboratory findings for PUB include hematemesis, melena, or hematochezia, alongside reduced hemoglobin or hematocrit levels. Endoscopy is essential for directly visualizing ulcerative lesions as the source of bleeding (Laine et al., 2021).

The following data were collected by reviewing the medical records of the patients: demographic information, vital signs, comorbidities, clinical presentation at admission, medications, endoscopic findings, and laboratory results. Additionally, four different risk scoring system scores were calculated for each patient.

The AIMS65 score was calculated by assigning one point for each of the following criteria: being over 65 years of age, serum albumin level below 3 g/dL, international normalized ratio (INR) value above 1.5, presence of mental status changes, and systolic blood pressure below 90 mmHg. The score ranges from 0 to 5, with higher scores being associated with an increased risk of mortality (Saltzman et al., 2011). The GBS is based on blood urea nitrogen level, hemoglobin level, systolic blood pressure, pulse rate, melena, syncope, and the presence of liver and heart disease. The score ranges from 0 to 23, where lower scores indicate low-risk patients, and higher scores identify those at risk of requiring intervention or at risk of mortality (Blatchford, Murray & Blatchford, 2000). The T-score is calculated based on the patient’s general condition, pulse rate, systolic blood pressure, hemoglobin level, and the number of comorbidities. Higher T-scores indicate a higher risk of mortality and rebleeding (Tammaro et al., 2014). The ABC score includes parameters such as age over 75, blood urea nitrogen level above 10 mmol/L, albumin level below 30 g/L, creatinine level above 150 µmol/L, mental status changes, comorbidities including liver cirrhosis and malignancy, and the American Society of Anesthesiologists (ASA) score. A higher ABC score is associated with an increased risk of mortality (Laursen et al., 2021).

The primary outcome of the study was in-hospital mortality, and the mortality status of patients was obtained through the hospital’s electronic database.

Analysis

Statistical analyses were conducted using SPSS for Windows (Version 29, Chicago, IL, USA) and MedCalc (Version 20.104, MedCalc Software Ltd., Ostend, Belgium). Continuous variables were expressed as mean ± standard deviation (SD) or median (interquartile range (IQR)), depending on data distribution. Categorical variables were presented as frequencies and percentages. The normality of continuous data was assessed using the Kolmogorov–Smirnov test, supplemented by histogram analysis to distinguish between parametric and non-parametric distributions. For between-group comparisons, independent samples t-tests were utilized for normally distributed continuous variables, while the Mann–Whitney U test was applied for non-normally distributed variables. Categorical variables were compared using the Chi-square test or Fisher’s exact test, as appropriate.

Receiver operating characteristic (ROC) curve analysis was employed to evaluate the diagnostic performance of the AIMS65, GBS, T-score, and ABC score in predicting in-hospital mortality. The area under the curve (AUROC) values were calculated with 95% confidence intervals (CIs) for each score. Pairwise comparisons of ROC curves were conducted using the DeLong test to assess differences between AUROC values. Sensitivity, specificity, and associated cutoff points were determined for each score based on the Youden index. A p-value of <0.05 was considered statistically significant for all analyses.

Results

Out of 356 patients initially reviewed, 41 were excluded due to incomplete data (n = 25) and alternative diagnoses (n = 16), leaving 315 patients for the final analysis. A total of 315 patients were included in the study, comprising 290 survivors (n = 290, 92.1%) and 25 deceased patients (n = 25, 7.9%). As presented in Table 1, the mean age of survivors (78.2 ± 8.2 years) was 3.7 years (95% CI [0.5–7.2]) less than that of the deceased group (81.9 ± 7.2 years) (p = 0.026). The proportion of males among survivors (n = 170, 58.6%) was not significantly different compared to the deceased group (n = 12, 48%) (p = 0.302).

Altered mental status was present in six deceased patients (24%), which was more frequent than among survivors (n = 7, 2.4%) (p < 0.001). The mean systolic blood pressure of survivors (124.2 ±  18.9 mmHg) was 13.9 mmHg (95% CI [6.1–21.6]) higher than that of the deceased group (110.3 ± 18.1 mmHg) (p < 0.001). The mean pulse rate was 96.2 ± 14.1 bpm for survivors, which was 6.2 bpm (95% CI [0.4–12.1]) lower than that of the deceased patients (102.4 ± 17.4 bpm) (p = 0.037).

The comorbidity profile, shown in Table 1, revealed that dementia was more prevalent in the deceased group (n = 9, 36%) compared to survivors (n = 25, 8.6%) (p < 0.001). However, no significant differences were found between survivors and deceased patients for coronary artery disease (p = 0.799), congestive heart failure (p = 0.469), stroke (p = 0.925), COPD (p = 0.795), diabetes mellitus (p = 0.574), liver disease (p = 0.603), and cancer (p = 0.182).

Table 2 describes clinical presentations and medication use. NSAID use was significantly more common among deceased patients (n = 9, 36%) compared to survivors (n = 31, 10.7%) (p < 0.001). There were no statistically significant differences in the presenting symptoms between the groups. Syncope was observed in nine survivors (3.1%) and two deceased patients (8%) (p = 0.214). Melena occurred in 254 survivors (87.6%) and 20 deceased patients (80%) (p = 0.279). Hematemesis was present in 71 survivors (24.5%) and 10 deceased patients (40%) (p = 0.089). Steroid use was observed in 4.8% of survivors (n = 14) compared to 12% of deceased patients (n = 3) (p = 0.128). Anticoagulant use was seen in 22.8% of survivors (n = 66) compared to 28% of deceased patients (n = 7) (p = 0.551), while antiplatelet use was reported in 38.6% of survivors (n = 112) compared to 48% of deceased patients (n = 12) (p = 0.357). Gastric ulcers were observed in 71.4% of survivors (n = 207) compared to 80% of deceased patients (n = 20) (p = 0.357). Duodenal ulcers were present in 28.6% of survivors (n = 83) compared to 20% of deceased patients (n = 5).

Table 1 Baseline characteristics and comorbidities.

Group	Variable	Survivor (n = 290)	Deceased (n = 25)	p	Mean Difference (95% CI)	
Demographics	Age (years)	78.2 ± 8.2	81.9 ± 7.2	0.026	3.7 [0.5–7.2]	
Sex (male)	170 (58.6%)	12 (48%)	0.302		
Vital signs	GCS	15 (15–15)	15 (14.5–15)			
Altered mental status	7 (2.4%)	6 (24%)	<0.001		
Systolic BP (mmHg)	124.2 ± 18.9	110.3 ± 18.1	<0.001	13.9 [6.1–21.6]	
Pulse (/min)	96.2 ± 14.1	102.4 ± 17.4	0.037	6.2 [0.4–12.1]	
Comorbidities	Coronary Artery Disease	120 (41.4%)	11 (44%)	0.799		
Congestive Heart Failure	63 (21.7%)	7 (28%)	0.469		
Stroke	33 (11.4%)	3 (12%)	0.925		
Dementia	25 (8.6%)	9 (36%)	<0.001		
COPD	30 (10.3%)	3 (12%)	0.795		
Diabetes Mellitus	85 (29.3%)	6 (24%)	0.574		
Liver Disease	10 (3.4%)	1 (4%)	0.603		
Cancer	32 (11%)	5 (20%)	0.182		
Notes.

GCS Glasgow Coma Scale

BP Blood Pressure

COPD Chronic Obstructive Pulmonary Disease

CI Confidence Interval

Table 2 Clinical presentation and medications.

Group	Variable	Survivor (n = 290)	Deceased (n = 25)	p	
Presenting symptoms	Syncope	9 (3.1%)	2 (8%)	0.214	
Melena	254 (87.6%)	20 (80%)	0.279	
Hematemesis	71 (24.5%)	10 (40%)	0.089	
Medications	Steroid	14 (4.8%)	3 (12%)	0.128	
Anticoagulants	66 (22.8%)	7 (28%)	0.551	
Antiplatelet	112 (38.6%)	12 (48%)	0.357	
NSAIDs	31 (10.7%)	9 (36%)	<0.001	
Bleeding source	Gastric Ulcer	207 (71.4%)	20 (80%)	0.357	
Duodenal Ulcer	83 (28.6%)	5 (20%)		
Notes.

NSAIDs, Nonsteroidal Anti-inflammatory Drugs.

Laboratory findings and prognostic scores are summarized in Table 3. The median hemoglobin level for survivors was 8.4 g/dL (IQR 6.4–10.1), which was not significantly different from that of the deceased group at 8.5 g/dL (IQR 5.9–10.2) (p = 0.868). The median BUN level was higher in deceased patients at 43.6 mg/dL (IQR 35.5–64.6) compared to survivors at 29 mg/dL (IQR 19.9–42.5) (p < 0.001). The median creatinine level was statistically significantly higher in deceased patients at 1.47 mg/dL (IQR 1–2.59) compared to survivors at 0.99 mg/dL (IQR 0.8–1.38) (p = 0.002). The median albumin level was statistically significantly lower in deceased patients at 29.1 g/L (IQR 24–32.3) compared to survivors at 34.1 g/L (IQR 30.1–38.1) (p < 0.001). The median INR was statistically significantly higher in deceased patients at 1.35 (IQR 1.15–1.59) compared to 0.98 (IQR 0.83–1.16) in survivors (p < 0.001).

Table 3 Laboratory findings and prognostic scores.

Group	Variable	Survivor (n = 290)	Deceased (n = 25)	p	
Laboratory values	Hemoglobin (g/dL)	8.4 (6.4–10.1)	8.5 (5.9–10.2)	0.868	
BUN (mg/dL)	29 (19.9–42.5)	43.6 (35.5–64.6)	<0.001	
Creatinine (mg/dL)	0.99 (0.8–1.38)	1.47 (1–2.59)	0.002	
Albumin (g/L)	34.1 (30.1–38.1)	29.1 (24–32.3)	<0.001	
INR	0.98 (0.83–1.16)	1.35 (1.15–1.59)	<0.001	
Prognostic score	AIMS65	1 (1–2)	2 (1.5–2.5)	<0.001	
Glasgow blatchford score	13 (10–14)	14 (10–15)	0.047	
T Score	7 (6–8)	7 (6.5–8)	0.295	
ABC Score	4 (3–5)	6 (4.5–7)	<0.001	
Outcome	LOS hospital (days)	5 (5–7)	5 (5–7)	0.449	
Notes.

BUN Blood Urea Nitrogen

INR International Normalized Ratio

LOS Length of Stay

AIMS65 Albumin, INR, Mental status, Systolic BP, Age ≥65 Score

ABC Score Age, Blood pressure, Consciousness Score

Prognostic scores indicated that the median AIMS65 score was statistically significantly lower in survivors at 1 (IQR 1–2) compared to deceased patients at 2 (IQR 1.5–2.5) (p < 0.001). The median Glasgow Blatchford Score was also statistically significantly lower in survivors at 13 (IQR 10–14) compared to 14 (IQR 10–15) in deceased patients (p = 0.047). The median T Score did not differ significantly between survivors and deceased patients, both groups having a score of 7 (IQR 6–8 for survivors; IQR 6.5–8 for deceased patients) (p = 0.295). The median ABC Score was statistically significantly lower in survivors at 4 (IQR 3–5) compared to 6 (IQR 4.5–7) in deceased patients (p < 0.001). The median length of hospital stay was 5 days (IQR 5–7) for both groups, with no statistically significant difference (p = 0.449).

Table 4 summarizes the AUROC, Youden index J, associated criterion, sensitivity, and specificity for each prognostic score. The AIMS65 score demonstrated an AUROC of 0.829 (95% CI [0.796–0.859]; p < 0.001), with a Youden index of 0.580 at a criterion of >1, yielding a sensitivity of 74.2% (95% CI [55.4–88.1]) and a specificity of 83.8% (95% CI [80.5–86.7]) (Fig. 1). The GBS showed an AUROC of 0.694 (95% CI [0.655–0.731]; p = 0.0001), with a Youden index of 0.362 at a criterion of >12, achieving a sensitivity of 74.2% (95% CI [55.4–88.1]) and a specificity of 62% (95% CI [57.9–66.1]). The T-score had an AUROC of 0.526 (95% CI [0.485–0.567]; p = 0.656), indicating no statistically significant discrimination capability. The ABC score demonstrated an AUROC of 0.775 (95% CI [0.739–0.808]; p = 0.001), with a Youden index of 0.407 at a criterion of >3, yielding a sensitivity of 77.4% (95% CI [58.9–90.4]) and a specificity of 63.3% (95% CI [59.1–67.3]).

Table 4 Prognostic score performance metrics.

Score	AUROC
(95% CI)	P	J	Associated criterion	Sensitivity (95% CI)	Specificity (95% CI)	
AIMS65	0.829
(0.796–0.859)	<0.001	0.580	>1	74.2
[55.4–88.1]	83.8
[80.5–86.7]	
GBS	0.694
(0.655–0.731)	0.0001	0.362	>12	74.2
[55.4–88.1]	62
[57.9–66.1]	
T-Score	0.526
(0.485–0.567)	0.656					
ABC Score	0.775
(0.739–0.808)	0.001	0.407	>3	77.4
[58.9–90.4]	63.3
[59.1–67.3]	
Notes.

AUROC Area Under the Receiver Operating Characteristic Curve

CI Confidence Interval

GBS Glasgow Blatchford Score

AIMS65 Albumin, INR, Mental Status, Systolic BP, Age ≥65 Score

ABC Score Age, Blood Pressure, Consciousness Score

J Youden Index

Figure 1 Receiver operating characteristic curves for AIMS65, Glasgow-Blatchford score, T-score, and ABC score in predicting in-hospital mortality in geriatric patients with peptic ulcer bleeding.

Pairwise comparisons of ROC curves were conducted to evaluate the relative performance of the scoring systems. The AUROC difference between the ABC score and AIMS65 was 0.0546 (SE 0.0481, 95% CI [−0.0397 to 0.149]; p = 0.2565), indicating no statistically significant difference. Similarly, the difference in AUROC between the ABC score and GBS was 0.0804 (SE 0.0588, 95% CI [−0.0348 to 0.196]; p = 0.1713), also showing no statistically significant difference. However, the comparison between AIMS65 and GBS revealed a statistically significant AUROC difference of 0.135 (SE 0.0587, 95% CI [0.0200–0.250]; p = 0.0214), indicating that AIMS65 was significantly better in discriminating in-hospital mortality compared to GBS.

Discussion

In this study, the effectiveness of four different risk scoring systems (AIMS65, GBS, T-score, and ABC score) in predicting in-hospital mortality in geriatric patients with PUB was compared. The results of the study concluded that the AIMS65 and ABC scores are stronger tools for predicting mortality in the geriatric population. In particular, the AIMS65 score demonstrated more successful outcomes in predicting mortality compared to other systems.

The diagnosis of UGIB in geriatric patients becomes more complex due to age-related physiological changes and accompanying comorbidities. With aging, the weakening of the gastric and intestinal mucosa can increase the frequency of gastrointestinal bleeding. In addition, these patients often use anticoagulants and antiplatelet medications, which are significant risk factors that can increase the severity of the bleeding. In geriatric patients, bleeding may present with more subtle and less obvious symptoms compared to younger patients, making it clinically challenging to detect the bleeding early. As hemodynamic compensation mechanisms weaken with age, symptoms like hypotension and shock often appear later, which can further delay the diagnosis process (Mahady et al., 2021; Xue et al., 2024).

The pathophysiology of UGIB is generally linked to the disruption of mucosal barriers, leading to peptic ulcers, erosions, or variceal bleeding. In elderly patients, this process becomes more complicated due to decreased mucosal blood flow, slower cellular regeneration, and a weakened immune system. The more severe progression of these pathophysiological processes in the geriatric population contributes to higher mortality rates. Therefore, early detection of UGIB in this patient group is of vital importance. Early diagnosis can help reduce both mortality and length of hospital stay. Risk scoring systems, particularly in geriatric patients, emerge as essential tools that assist in making rapid clinical decisions early in the disease course. Accurate and prompt risk assessment enables the identification of high-risk patients and directs them to advanced treatment processes such as intensive care or emergency endoscopy, thereby improving outcomes (Lanas et al., 2009; Jairath et al., 2012; Kim, Choi & Shin, 2019; Tatlıparmak et al., 2022).

Our finding that the AIMS65 score is a strong predictor of mortality in geriatric patients is consistent with the current literature. AIMS65 has been frequently emphasized in the literature as an effective tool for predicting mortality in older age groups. For example, in the study by Ak & Hökenek (2021), it was reported that AIMS65 is a better predictor of mortality compared to the Glasgow Blatchford Score (GBS), with an AUC value of 0.706 for AIMS65 and 0.542 for GBS. Similarly, in the study by Lu, Zhang & Chen (2020), the AUC value for AIMS65 was reported as 0.955, while the AUC for GBS was 0.882, demonstrating the superiority of AIMS65 over GBS. A decline in admissions for bleeding peptic ulcers between 2017 and 2021 has been observed, with mortality rates remaining stable despite the changes in admission patterns (Cazacu et al., 2024). These findings highlight the importance of effective triage and risk stratification tools like AIMS65, which can guide timely clinical decisions and optimize patient outcomes. However, the limited ability of GBS to predict mortality in our study suggests that its accuracy in the geriatric population may be lower. While GBS has shown strong predictive power for the need for interventional procedures, its performance in predicting mortality has been reported as weaker in previous studies as (Laursen et al., 2015).

One of the important findings of this study is that the ABC score performed similarly to the AIMS65 score. In the literature, the ABC score has been noted as a powerful tool for predicting mortality, especially in elderly patients with multiple comorbidities. In a multicenter study by Laursen et al. (2021) the ABC score was found to outperform other existing scoring systems in predicting mortality in both upper and lower gastrointestinal bleeds and was more accurate in identifying low-risk patients. Similarly, in the study by Li et al. (2022) on geriatric patients, the ABC score was noted to have the highest accuracy in predicting mortality, which supports the findings of our study.

On the other hand, the lower-than-expected performance of the T-score in predicting mortality in this study suggests that the use of this scoring system may be limited in the geriatric population. A similar finding was reported in the study by Ebrahimi Bakhtavar et al. (2017), where the T-score was found to have lower accuracy compared to other scoring systems, particularly in the geriatric population. These findings suggest that the T-score may need to be re-evaluated as an appropriate predictive tool for the geriatric patient group.

Limitations

This study has several limitations. First, as a retrospective study, it relied on the accuracy and completeness of medical records, which may have introduced bias due to missing or unreported medical history or symptoms during emergency department presentation. Additionally, the study was conducted at a single center, which may limit the generalizability of the results. Larger, multicenter studies are needed to confirm the findings in different clinical settings. The exclusion of patients under 65 years of age limits the ability to generalize the results to younger populations. Finally, while we evaluated four scoring systems, other tools not included in this study may also provide valuable insights into mortality prediction in geriatric patients with upper gastrointestinal bleeding.

Conclusions

This study demonstrates that the AIMS65 and ABC scoring systems are more effective in predicting in-hospital mortality in geriatric patients with PUB compared to the GBS and T-score. These findings suggest that incorporating these tools into clinical practice could improve decision-making processes in managing high-risk elderly patients. Future prospective studies are warranted to validate these results and explore their applicability in broader clinical settings.

Supplemental Information

Supplemental Information 1 Dataset

All variables and data points necessary for replicating the results, such as demographic information, clinical parameters, laboratory values, and calculated scores for the AIMS65, GBS, T-score, and ABC scoring systems. Use SPSS to open this file.

Supplemental Information 2 Codebook to convert numbers to their respective factors in dataset categorical data

Additional Information and Declarations

Competing Interests

Author Contributions

Ethics

Data Availability

The authors declare there are no competing interests.

Omerul Faruk Aydin conceived and designed the experiments, performed the experiments, prepared figures and/or tables, authored or reviewed drafts of the article, and approved the final draft.

Ali Cankut Tatlıparmak conceived and designed the experiments, performed the experiments, analyzed the data, prepared figures and/or tables, authored or reviewed drafts of the article, and approved the final draft.

The following information was supplied relating to ethical approvals (i.e., approving body and any reference numbers):

The Yeni Yüzyıl University Clinical Research Ethics Committee approved the study (Date: 04.11.2024, Decision No: 2024/11-1358).

The following information was supplied regarding data availability:

Raw data is available in the Supplemental Files.

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
