# Peer review of "Predicting mortality in geriatric patients with peptic ulcer bleeding: a retrospective comparative study of four scoring systems"

_PeerJ, doi:10.7717/peerj.19090_

## Round 0.1 · original submission · Major Revisions

Please address the comments supplied.

Reviewer 1 ·

Basic reporting

The manuscript is clear and well-structured, written in professional English, with relevant and current references. Figures and tables are appropriate and support the manuscript effectively.

Suggested Revision: In the "Results" section, the manuscript currently does not specify the total number of patients initially considered for inclusion and how many were excluded based on the stated exclusion criteria. Adding a brief explanation of the total cohort and exclusions (e.g., "Out of X patients initially reviewed, Y were excluded due to [specific reasons]") would provide greater clarity and improve transparency.

Experimental design

The research question is well-defined, and the study design is appropriate to answer the research question. The methodology is described in sufficient detail for replication.

No additional comments.

Validity of the findings

The findings are statistically sound, well-supported by the data, and appropriately discussed.

No additional comments.

Additional comments

This is a well-conducted and valuable study. Addressing the minor revision suggested above will enhance the manuscript's clarity and adherence to reporting standards.

Reviewer 2 ·

Basic reporting

No comment

Experimental design

The study design is appropriate and effectively addresses the research question. The retrospective cohort approach is well-suited to evaluate the predictive performance of scoring systems in geriatric patients.

Suggested Revisions:

The authors define geriatric patients as those aged 65 years and older but do not provide a rationale or reference for this threshold. It is recommended to include a justification or reference to support this definition, ensuring consistency with existing literature or clinical guidelines.
The definition of upper gastrointestinal bleeding (UGIB) is mentioned but could benefit from further clarification. Specifically, it would be helpful to include the diagnostic criteria or key clinical and endoscopic findings used to confirm UGIB, enhancing the reproducibility and understanding of the study.

Validity of the findings

No comment

Additional comments

This is a valuable study addressing an important clinical question. The manuscript is well-structured and provides meaningful insights into the predictive performance of scoring systems in geriatric patients with UGIB.

Suggested Revisions:

As noted in the "Experimental Design" section, please consider elaborating on the rationale for selecting 65 years as the threshold for geriatric patients and provide a reference to support this definition.
Additionally, refining the definition of upper gastrointestinal bleeding (UGIB) by including specific diagnostic criteria or key findings will enhance the clarity and reproducibility of the study.
Addressing these points will improve the transparency and overall quality of the manuscript.

·

Basic reporting

1- Clear, unambiguous, professional English language is used throughout.
The paper used clear and professional English.
2- Intro & background to show context.
The introduction contains relevant information for the objectives of the paper.
3- Literature well referenced & relevant.
The literature included in the Reference section is relevant. However, the publication year is missing in some references, the citations are in alphabetical order, and not in the order of the citation in the paper, at the end of the Discussion section some references are also cited in rectangular brackets. Consistency must be done to improve the quality of the paper.
4- Structure conforms to PeerJ standards, discipline norms, or improved for clarity.
The structure of the paper conforms to PeerJ standards.
5- Figures are relevant, high quality, well labeled & described.
The figure is relevant, of high quality, and well-labeled.
6- Raw data supplied (see PeerJ policy).
The raw data were supplied.

Experimental design

1- Original primary research within the Scope of the journal.
The original primary research fits PeerJ's scope.
2- Research question well defined, relevant & meaningful. It is stated how the research fills an identified knowledge gap.
The research question was well-defined and relevant. The accuracy of prognostic scores in geriatric patients with UGIB is useful in order to better estimate the prognosis in this special category of patients.
However, the patient group contains ONLY PEPTIC ULCER BLEEDING CASES (see also Table 2). In this setting, the sections TITLE, ABSTRACT, MATERIAL and METHOS, DISCUSSIONS, and CONCLUSION must be amended to replace UGIB with PEPTIC ULCER BLEEDING.
3- Rigorous investigation performed to a high technical & ethical standard.
The investigation performed was rigorous, with statistical accuracy, and conform to the ethical standards, with the same observation regarding PEPTIC ULCER BLEEDING instead of UGIB.
4- Methods described with sufficient detail & information to replicate.
The methods were described with sufficient detail, with the same observation regarding PEPTIC ULCER BLEEDING instead of UGIB.

Validity of the findings

VALIDITY OF THE FINDINGS
All underlying data have been provided; they are robust, statistically sound, & controlled.
Conclusions are linked to the original research question and are limited to supporting results. However, the same observation regarding PEPTIC ULCER BLEEDING instead of UGIB is applicable.

Additional comments

In Lines 74-75, I consider that the written consent by the patients is mandatory in retrospective studies only for patient data protection (which is not breached in the current paper).
In Line 98, the ABC score includes not only mental status change but also other comorbidities (Liver cirrhosis, malignancy) and ASA Score.
In Table 1, the difference in age between survivors and deceased patients is 3.7 and in line 122 from the text is 3.8 (possibly related to the rounding effect), and the same observation for the pulse.
In Table 1, the prevalence of CKD is similar between survivors and deceased, but in Table 3 the median creatinine level in deceased patients is 1.35 (which, given the age above 65 in all patients, may suggest a higher percentage of CKD in deceased cases). A reassessment of data may be useful in order to help clarify this observation.

---

## Round 0.2 · accepted · Accept

Dear Authors

Thank you for making all the changes recommended by the referees.

Congratulations!

Reviewer 1 ·

Basic reporting

no comment

Experimental design

no comment

Validity of the findings

no comment

Additional comments

Thank you for carefully addressing the suggested revisions. The changes made have significantly improved the clarity and quality of the manuscript.

Reviewer 2 ·

Basic reporting

no comment

Experimental design

no comment

Validity of the findings

no comment

Additional comments

Thank you for addressing the suggested revisions thoroughly.

·

Basic reporting

No comment.

Experimental design

No comment

Validity of the findings

No comment

Additional comments

The authors submit a clear response to all observations.